# Spinal muscular atrophy genetic epidemiology and the case for premarital genomic screening in Arab populations
Fatma Rabea[1,2], Maha El Naofal[2], Ikram Chekroun[1], Mona Khalaf[3], Nuha Al Zaabi[4], Khawla AlZaabi[5], Mahmoud ElHalik [6], Swarup Dash[6], Yaser El Saba[7], Azhari Ali[8], Smitha Abraham[9], Khansa Fathi[10], Jwan Shekhy[11], Saad G. Aswad[12], Haitham Elbashir[13], Fowzan Alkuraya [14], Tom Loney [1], Alawi Alsheikh-Ali[1], Abdulla Al Khayat[15] & Ahmad Abou Tayoun[1,2] ✉

## Abstract

**Background** Spinal muscular atrophy (SMA) is a fatal autosomal recessive disorder for which several treatment options, including a gene therapy, have become available. SMA incidence has not been well-characterized in most Arab countries where rates of consanguinity are high. Understanding SMA disease epidemiology has important implications for screening, prevention, and treatment in those populations.

**Methods** We perform SMA diagnostic testing in a clinical multi-national patient cohort (N = 171) referred for hypotonia and/or muscle weakness. In addition, we carry out genetic newborn screening for SMA on 1502 healthy Emirati newborns to estimate the carrier frequency and incidence of the disease in the United Arab Emirates.

**Results** Patients referred for SMA genetic testing are mostly Arabs (82%) representing 18 countries. The overall diagnostic yield is 33.9%, which is higher (>50%) for certain nationalities. Most patients (71%) has two *SMN2* copies and earlier disease onset. For the first time, we estimate SMA carrier frequency (1.3%) and incidence of the disease (1 in 7122 live births) in the United Arab Emirates. Using birth and marriage rates in two Arab populations (United Arab Emirates and Saudi Arabia), as well as disease incidence in both countries, we show that, besides preventing new cases, premarital genetic screening could potentially result in around $8 to $324 million annual cost savings, respectively, relative to postnatal treatment.

**Conclusions** The SMA carrier frequency and incidence we document suggests high potential benefit for universal implementation of premarital genomic screening for a wide range of recessive disorders in Arab populations.

## Plain language summary

The occurrence of spinal muscular atrophy, a fatal genetic nerve and muscle disease, has been poorly studied in most Arab countries. Individuals who carry a single mutated gene copy (carriers) may be more likely to marry other carriers in regions where marriage rates amongst relatives, who share similar genetics, are high. Here we report the results of a newborn testing program for this disease in 1502 Emiratis and calculate the presence of carriers (1/79) and occurrence of disease (1/7122) in this population. Using this new information along with the annual birth and marriage rates in the United Arab Emirates and Saudi Arabia, we make the case that premarital genomic screening (carrier testing) is the best way to prevent this and other similarly inherited disorders in the Arab population.

Spinal muscular atrophy (SMA) is a rare genetic neuromuscular disorder characterized by irreversible loss of alpha motor neurons in the spinal cord and lower brainstem resulting in progressive muscle atrophy and weakness[1]. Its clinical manifestation is highly variable, ranging in onset from intrauterine to adulthood, and in severity from minor muscle weakness to severe paralysis and death. Accordingly, SMA has been classified into four primary subtypes: type I, type II, type III and type IV. Type I, known as infantile SMA, is the most common and severe form of the disease, seen in ~50%[2] of affected patients and has an onset before

6 months of age. If untreated, death generally occurs within the first 2 years of life due to respiratory failure[3,4].

SMA is an autosomal recessive disorder caused by biallelic pathogenic variants in the telomeric survival motor neuron 1 (*SMN1*) gene located on chromosome 5q13[5]. Most affected patients (92%) harbor a homozygous deletion of *SMN1* exon 7, while a small subset (4%) are compound heterozygotes for exon 7 deletion on one allele coupled with a small intragenic variant on the other allele[6]. In rare occasions (4%), SMA can be due to over 30 genes identified, other than *SMN1*[3,6,7]. Though an almost identical

---

homolog to *SMN1*, *SMN2* can only partially compensate for *SMN1* loss. Increase in *SMN2* copy number has been associated with milder forms of SMA and, therefore, has prognostic value. Of patients with 4 copies of *SMN2*, 1% of patients had SMA type I, 11% had SMA type II and 88% had SMA type III or type IV[4,7].

SMA is globally the second most common fatal autosomal recessive disorder. Based on available data in populations of European ancestry, SMA highest incidence and carrier frequency are around 1 in 8000 and 1 in 45, respectively[6,7]. These figures are expected to be even higher in regions with high rates of consanguinity, particularly in the Middle East where the incidence has been predicted to range between 10 and 193 in 100,000 live births, almost 40-fold higher than in Western populations[8]. However, there is a lack of data to accurately estimate SMA epidemiology across Arab populations. Here we estimate the carrier frequency and incidence of SMA in the United Arab Emirates (UAE) and, using this information, we show that premarital genomic screening is the most efficient public health measure for SMA, and other recessive diseases, prevention in Middle Eastern population.

## Methods
### Study design and participants
**Clinical patient cohort.** Pediatric patients (age < 18 years old) were referred for diagnostic SMA testing at the Genomics Centre in Al Jalila Childrens' Specialty Hospital between June 2019 and August 2023. Al Jalila Children's is a tertiary pediatric center in the UAE and is the main referral center for pediatric patients with rare diseases across the UAE. Peripheral blood samples were obtained from patients suspected with SMA and clinical data was provided by the referring physician either through the electronic medical record, for internal patients, or on requisition forms, for external patients. The de-identified reporting of clinical cohort in this study was approved by the Dubai Healthcare Authority Research Ethics Committee (DSREC-07/203_06), which determined that this study meets the exemption criteria with a waiver of informed consent since all data were de-identified. A subset of this clinical cohort was previously published[9], though in addition to increasing cohort size, this current study includes additional information on age ranges, diagnostic yield stratified by nationality, and *SMN2* copy number status. Diagnostic yield was calculated by dividing the number of patients positive for the biallelic deletion of the *SMN1* gene by the total number of tested patients.

**SMA genetic screening initiative in Emirati Newborn.** A network of ten local maternity hospitals across the UAE was established to conduct the country's first large scale representative genetic newborn screening study for SMA. Recruitment and consenting of Emirati newborns' (age ≤ 3 months) families were carried out by neonatologists/pediatricians at each hospital. Peripheral blood samples were obtained from each participant and clinical data were provided by the recruiting physician

through an electronic collection data sheet. Samples were transported to Al Jalila Genomics Centre where SMA testing was performed. The study was approved by the Dubai Health Authority Scientific Research Ethics Committee (DSREC-06/2021_20), Ministry of Health and Prevention Research Ethics Committee (MOHAP/DXB-REC/SOO/No.81/2021) and Abu Dhabi Health Research and Technology Ethics Committee (DOH/CVDC/2022/1626). Parents or legal guardians of all recruited newborns provided written informed consent.

### DNA extraction
Genomic DNA was extracted from peripheral whole blood using the QIAsymphony DSP DNA Kit (Qiagen, Hilden, Germany) and QIAsymphony automated nucleic acid extraction instrument, according to the manufacturer's instructions.

### SMA copy number analysis
*SMN1* and *SMN2* copy numbers were determined by Digital droplet PCR (ddPCR) technology using predesigned proprietary ddPCR assay kits for *SMN1* (Catalog No: 186-3500, Bio-Rad) and *SMN2* (Catalog No: 186-3503, Bio-Rad). The predesigned Bio-Rad assays include reagents to detect both the target gene (*SMN1* or *SMN2*) and an internal reference gene (*RPP30*). In addition, experimental controls—0 copy, 1 copy and 2 copy controls for *SMN1*, and 2 copy, 3 copy and 4 copy controls for *SMN2*—were included along with a no template control.

Data analysis was performed using QuantaSoft version 1.7.4.0917 (Bio-Rad) to determine the copy number variation (CNV)[10,11] (Fig. 1). *SMN2* copy number was determined only for SMA-positive cases with a homozygous *SMN1* exon 7 deletion to assess disease severity and prognosis.

### Reporting summary
Further information on research design is available in the Nature Portfolio Reporting Summary linked to this article.

## Results and discussion
### SMA patient distribution and diagnostic yield in a multi-arab population
Between June 2019 and August 2023, 171 pediatric patients (51% females; mostly infants) (Fig. 2A) presenting with hypotonia and/or muscle weakness were referred to Al Jalila Genomics Centre for diagnostic SMA testing. Patients, representing 18 countries, were mostly Arabs (82%) with the majority from Saudi Arabia (29%), UAE (25%) and Iraq (10%) (Fig. 2B).

Testing was positive in 58 cases, making up an overall diagnostic yield of 33.9% (Fig. 2C), which varied by nationality (Fig. 2B); yield was significantly higher among patients from Iraq (53%) relative to the UAE (19%)

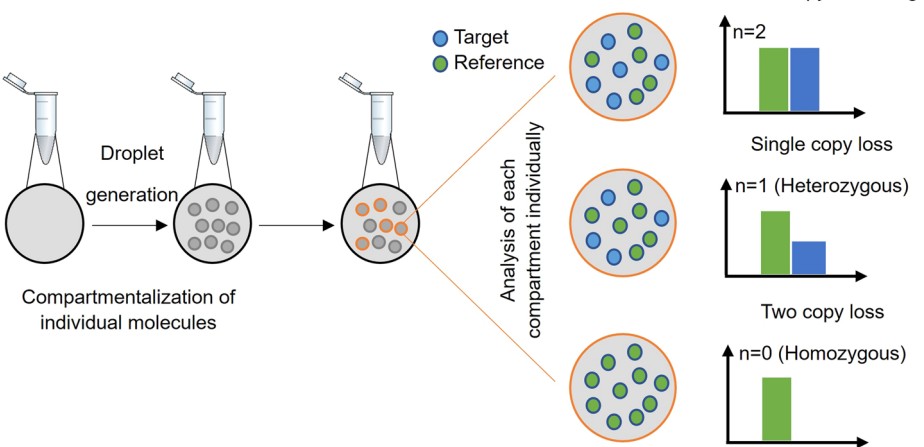

**Fig. 1 | Droplet Digital PCR workflow for *SMN1* and *SMN2* copy number analysis.** Probes specific to either *SMN1* or *SMN2* gene (Target, blue) or a reference gene (*RPP30*, green) were used. The number of target droplets (blue) were compared to that of reference droplets (green) to assess the copy number status of *SMN1* or *SMN2* (as illustrated).

**Fig. 2 | Spinal muscular atrophy patients' demographics and genetic findings. A** Distribution of patients by age and gender. **B** Distribution of patients ($n = 171$) and diagnostic yield by nationality. **C** Overall diagnostic yield. **D** *SMN2* copy number status in spinal muscular atrophy patients of different ages. **E** Testing turnaround time.

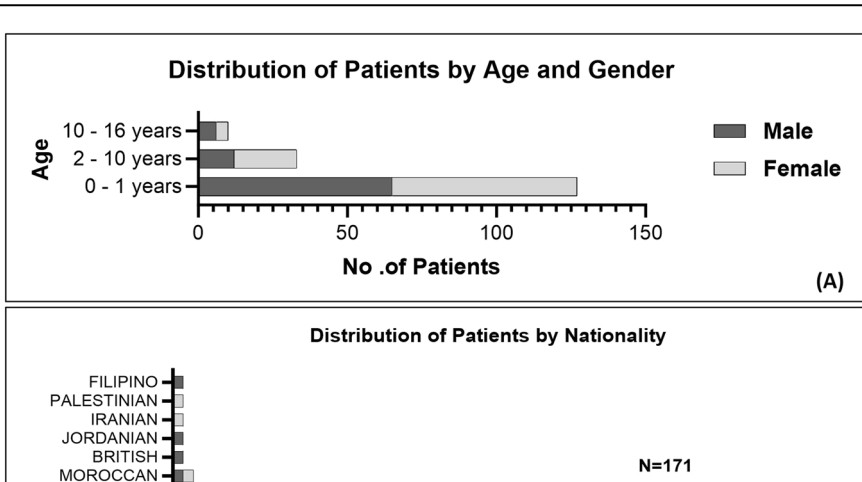

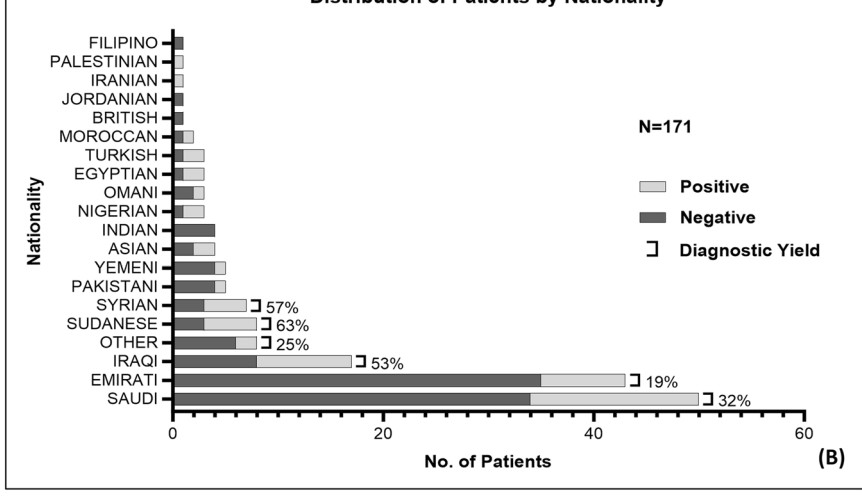

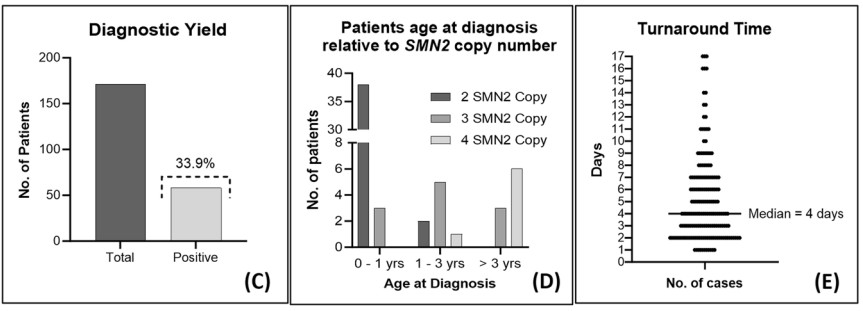

($P = 0.012$; Fisher Exact Test). Determination of *SMN2* copy number in SMA positive cases showed that the majority (71%) had 2 *SMN2* copies, consistent with an earlier onset (Fig. 2D) and a more likely severe phenotype (SMA type I)[7,12]. Average test turnaround time was 5 working days, with more than half (56%) receiving results in ≤4 working days (Fig. 2E).

## SMA epidemiology in the United Arab Emirates

We attempted to estimate the incidence of SMA in the UAE using the number of Emirati SMA patients born in a given year, as diagnosed by our laboratory during this period, divided by total annual live births in this population (which was 31,064 in 2021)[13]. However, since not all SMA diagnostic testing in the UAE is referred to our laboratory and due to the lack of national SMA registry, it is likely that this clinical-based incidence estimation to be inaccurate.

To gain better insight into SMA epidemiology in the UAE, we genetically screened a total of 1502 newborns (51% males, 49% females) recruited from 10 local maternity hospitals across the country (Fig. 3A). Newborns aged ≤3 months and were generally healthy with no family history of SMA. Results from genetic screening revealed that 19 newborns (47% females, 53% males) were carriers for *SMN1* deletion (Fig. 3B), indicating a carrier frequency of 1 in 79 (1.3%) and an SMA allele frequency (q) of 0.0063 (Table 1). None of the newborns tested had a homozygous *SMN1* deletion.

Assuming Hardy–Weinberg equilibrium, disease burden or incidence ($q^2$) can be calculated as 1 in 25,000 individuals. However, considering the non-random mating in this population, a more appropriate estimate for incidence would be q x f[14], where F represents the coefficient of inbreeding in the Emirati population ($F = 0.0222$)[15], resulting in an incidence of 1 in 7122 individuals (Table 1).

It is important to note that this is likely an underestimate since (1) our testing method only detects deletions that account for 92% of cases, while (2) this method does not distinguish silent carriers, who possess two copies of the *SMN1* gene on one allele (0 + 2), from normal noncarriers (1 + 1).

**Table 1 | SMA genetic epidemiology in two Arab populations based on screening studies**

| Country | UAE* | KSA[17] |
|---|---|---|
| **Sample size (N)** | 1502 | 4198 |
| **Carrier frequency** = number of carriers divided by total number of participants (N) | 1:79 (1.3%) | 1:39 (2.6%) |
| **Allele frequency (q)** = number of carriers divided by total number of chromosomes (N × 2) | 0.0063 | 0.0128 |
| **Coefficient of inbreeding (F)** | 0.0222[14] | 0.0241[14,18] |
| **Estimated incidence** = q x F | 1:7122 | 1:3192 |

*Data from this study; *UAE* United Arab Emirates, *KSA* Kingdom of Saudi Arabia.

**Fig. 3 | Newborn recruitment sites and SMA genetic screening results. A** Recruiting sites throughout the UAE. **B** Distribution of *SMN1* copy number in tested Emirati newborns (*N* = 1502). EHS Emirates Health Services, DH Dubai Health, DOH Department of Health.

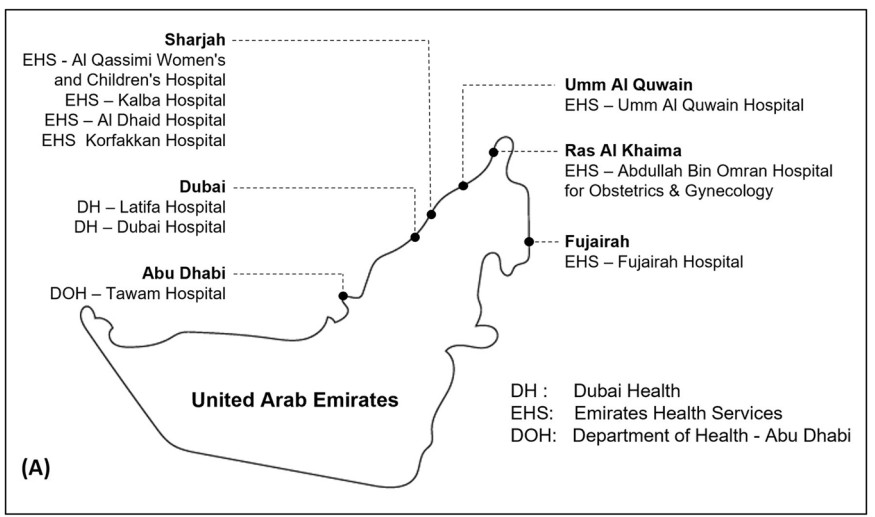

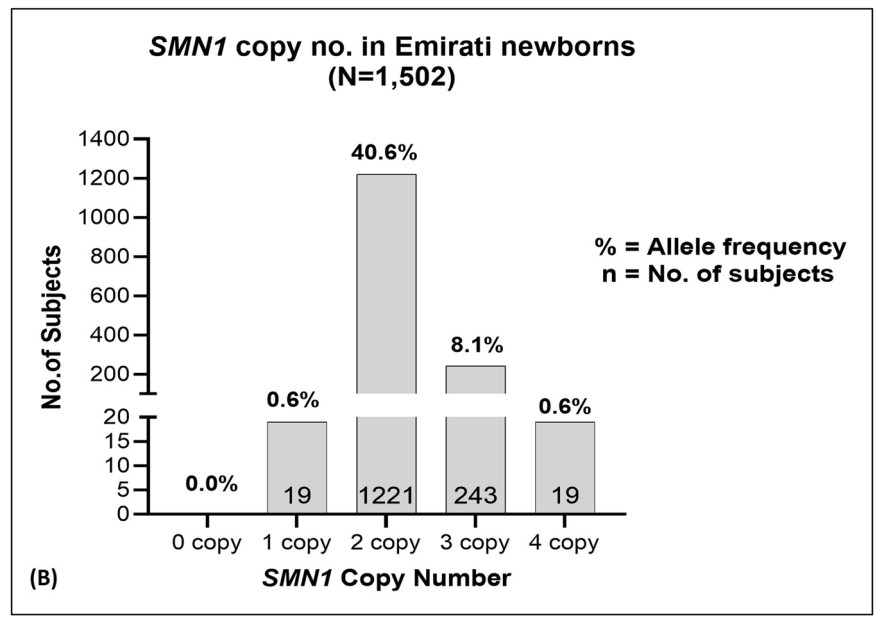

## SMA epidemiology in other Arab populations

Genetic newborn screening studies have been limited in most Arab countries. A study from Morocco reported a carrier frequency of 1 in 25 and a prevalence of 1:1900, though this was based on a small sample of 150 Moroccan newborns[16]. While this work was in preparation, a study was published whereby screening 4189 normal Saudi volunteers determined a carrier frequency of 1 in 38[17]. Using this SMA allele frequency, and the coefficient of inbreeding in Saudi Arabia[18], we estimated SMA incidence in this country to be 1 in 3192 (Table 1), which is consistent with that estimated by authors in this study (32 in 100,000 or 1 in 3125 births)[17]. Those estimates suggest the SMA carrier frequency and incidence in Saudi Arabia are both higher than those estimated in the UAE; findings which might be consistent with the observed higher SMA diagnostic trend in Saudi patients compared to those in the UAE (Fig. 2B).

## The case for premarital genomic screening

The tragic loss of a baby to SMA in Australia has fueled a large-scale reproductive carrier screening program for 750 genetic conditions ("Mackenzie's Mission") in Australia[19]. The estimated SMA incidence in both Arab countries studied here is higher than that in other populations worldwide[7], a finding which is consistent with the expectedly higher incidence of recessive disorders in Arab populations, suggesting that premarital genetic screening is likely to be an effective preventive measure in those populations.

Based on the aforementioned estimate of SMA incidence in the UAE (Table 1) and the average annual Emirati birth rate, we conservatively anticipate that 4 births per year will be affected with SMA, necessitating gene therapy at a total cost of $8.4 million ($2.1 million per patient; other long term treatment options, such as Spinraza, have a higher cost). However, if SMA premarital screening is implemented, the projected total cost of screening Emirati marriages would be $0.49 million annually (Table 2). Couples at risk (*N* = 4) will have the option for pre-implementation genetic testing (PGT) and in vitro fertilization (IVF) at a cost of up to $30,000 per couple[20] (Table 2).

Similarly, average annual birth count among Saudi Nationals (18.8 million) is 511,000[21,22]. Therefore, it is expected that 160 births will be affected with SMA per year, requiring $336 million for treatment. However, the cost of SMA premarital screening for the annual 141,000 Saudi marriages in KSA[22], would be around $7 million, along with a total cost of $4.8 million in PGT/IVF for at risk couples (Table 2).

**Table 2 | Estimates of SMA births in UAE and KSA and the associated cost of postnatal treatment versus premarital screening**

|  | United Arab Emirates | Saudi Arabia |
|---|---|---|
| Annual births[13,21,22] | 33,625 | 511,000 |
| SMA livebirths per year[*] | 4 | 160 |
| Cost of gene therapy[#] | $8.4 million | $336 million |
| Annual marriage rate[13,21,22] | 9875 | 141,000 |
| Cost of Premarital screening[^] | $0.49 million | $7 million |
| Cost of PGT/IVF[$] | $120,000 | $4.8 million |
| Cost reduction[&] | $7.79 million (14-fold) | $324.2 million (28-fold) |

[*]Calculated by multiplying incidence in Table 1 by annual births in each country, rounded to lowest integer.
[#]$2.1 million dollar.
[^]Calculated by multiplying cost of SMA testing ($50) per couple.
[$]Calculated by multiplying the number of couples expected to have an SMA birth and requiring PGT/IVF (pre-implementation genetic testing/in vitro fertilization) by cost of PGT/IVF (30,000 USD[20]).
[&]Calculated by subtracting cost of premarital screening and PGT/IVF from total cost of gene therapy.
*UAE* United Arab Emirates, *KSA* Kingdom of Saudi Arabia.

We estimate the disease burden or incidence of SMA in the UAE. Using data from the UAE (this study) and Saudi Arabia[17], we demonstrate that, in addition to potentially preventing new SMA cases, premarital screening and subsequent PGT/IVF for at-risk couples, can be a highly cost-effective measure from a public health standpoint, leading to an expected 14 to 28-fold cost reduction compared to postnatal disease treatment. Interestingly, the cost savings due to premarital screening for this one disease (~$8 million in UAE and $324 million in KSA) is sufficient to fund more comprehensive premarital genomic screening programs encompassing hundreds of recessive disorders in both countries. We strongly advocate for the implementation of comprehensive genomic premarital screening in Arab populations with similarly high consanguinity rates.

## Data availability
All data is included in the manuscript. Source data used to generate figures can be found in Supplementary Data 1.

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

## Acknowledgements

The spinal muscular atrophy pilot study is being supported by an unrestricted medical grant from Novartis/AveXis. We would like to thank Haneen Farajallah, Najeeb Muringakodan, Hanan Alraeesi, Saleh Ali, Adel Salem and Fatima AlKaabi from Emirates Health Services for supporting their respective teams in this project.

## Author contributions

A.A.T. conceived study and obtained funding. M.K., N.A., K.A., M.E., S.D., Y.S., A.A., S.A., K.F., J.S., and S.G.A. recruited and consented families for SMA genetic screening. F.R., M.E., I.C. and A.A.T. performed all analysis. A.A.T., F.A. and A.A.K. obtained ethical approvals. F.A., T.L. and A.A.A. helped with epidemiological analysis. F.R. and A.A.T. wrote first manuscript draft. All authors, including H.E., edited and approved the final draft.

## Competing interests
Authors declare no competing interests. Dr. Ahmad Abou Tayoun is an Editorial Board Member for Communications Medicine but was not involved in the editorial review or peer review, nor in the decision to publish this article.

## Additional information

[1]College of Medicine, Mohammed Bin Rashid University of Medicine and Health Sciences, Dubai Health, Dubai, UAE. [2]Al Jalila Genomics Center of Excellence, Al Jalila Children's Specialty Hospital, Dubai Health, Dubai, UAE. [3]Neonatology Department, Al Qassimi Women's & Children's Hospital, Emirates Health Services, Sharjah, UAE. [4]Pediatric Department, Fujairah Hospital, Emirates Health Services, Fujairah, UAE. [5]Pediatric Department, Kalba Hospital, Emirates Health Services, Sharjah, UAE. [6]Neonatal Section, Latifa Women & Children Hospital, Dubai Health, Dubai, UAE. [7]Department of Neonatology, Dubai Hospital, Dubai Health, Dubai, UAE. [8]Neonatology Department, Umm Al Quwain Hospital, Emirates Health Services, Umm Al Quwain, UAE. [9]Department of Neonatology, Abdullah Bin Omran Hospital, Emirates Health Services, Ras Al Khaimah, UAE. [10]Neonatology Department, Al Dhaid Hospital, Emirates Health Services, Sharjah, UAE. [11]Neonatology Department, Khorfakkan Hospital, Emirates Health Services, Sharjah, UAE. [12]General-Obs/Gyno Clinic, Tawam Hospital, Al Ain City, Abu Dhabi, UAE. [13]Neurosceince Center of Excellence, Al Jalila Children's Specialty Hospital, Dubai Health, Dubai, UAE. [14]Departement of Translational Genomics, Center for Genomic Medicine, King Faisal Specialist Hospital and Research Center, Riyadh, Saudi Arabia. [15]Al Jalila Children's Specialty Hospital, Dubai Health, Dubai, UAE. ✉e-mail: Ahmad.Tayoun@dubaihealth.ae

