## [Peer Review File · Communications Medicine]

Reviewers' comments:

Reviewer #1 (Remarks to the Author):

Dear Editor,

I read with interest the case report entitled (Spinal muscular atrophy genetic epidemiology and the case for premarital 2 genomic screening in Arab populations) reported by Dr. Abou-Tayoun and his colleagues. They did SMA diagnostic testing in a clinical multi-national patient cohort referred for hypotonia and/or muscle weakness. In addition, they carried out genetic newborn screening for SMA on 1,252 healthy Emirati newborns to estimate the carrier frequency and incidence of the disease in the United Arab Emirates.

In line 286 (Al-Jalila Children's is the only tertiary pediatric center in the UAE and is 287 the main referral center for pediatric patients with rare diseases across the UAE), just to modify this sentence as there are 2 other centers dealing with rare diseases cross UAE. Suggest to write:

" Al- Jalila Children's is one of tertiary pediatric centers in the UAE dealing with pediatric patients with rare diseases across the UAE".

Figure 3, without label of A and B.

In SMA Epidemiology in other Arab populations part. It will be great to document why Saudi Arabia chosen to be compared with UAE.

This study supports the previously reported by Hannah V. Wilson, Flavia M. Cantarutti, Islam Eltantawy, Alia A. Aldhaheeri, Yousuf M. Naqvi, Amrit Sadani and Muna M. Al Safi. In Real-World Evidence – The Clinical Burden of Spinal Muscular Atrophy (SMA) in Abu Dhabi, UAE. In which they reported all things related to SMA considered, the clinical burden of SMA in Abu Dhabi is significant, as reported worldwide. Thus the need for screening and prevention are mandated.

Reviewer #2 (Remarks to the Author):

This report is well written and organized. Authors describe epidemiology of SMA in Arab population of UAE. Their conclusion is well supported by statistical analysis. They report that incidence of SMA in UAE is higher due to consanguinity. They state that screening for carrier status may reduce incidence of SMA in population and therefore can have a significant economic impact by reducing healthcare spending on treatment of SMA.

I would recommend this manuscript for publication.

Reviewer #3 (Remarks to the Author):

The authors conducted two surveys. The first is an investigation of *SMN1* and *SMN2* copy numbers in pediatric patients with hypotonia or/and muscle weakness in the UAE. The second is a newborn screening survey for SMA in the UAE. There are few similar reports in the UAE, and this is considered to be a valuable report. On the other hand, I have some major concerns about this paper.

1. Estimated incidence

Of the 1,252 newborns, 19 had one copy of *SMN1* (1.5%). If 19 newborns are carriers, then $19/1,252 = 1$ in 65.9. As the authors write, the incidence (q^2) of SMA calculated from this is 1 in

17,368 (65.9*65.9*4) individuals. The authors used the F value to conclude that it was 1 in 5,990 individuals. This is nearly three times more frequent than q2. As an SMA expert, I don't understand this process. Therefore, it is important to explain in an easy-to-understand way to the reader the process and reason why the incidence has tripled by using the F value. "For the first time, we estimate SMA carrier frequency (1.5%) and incidence of the disease (1 in 5,990 live births) in the United Arab Emirates" in the abstract should be revised. I think the incidence of using q2 should be specified in the abstract. The incidence of using the F value in the abstract should be deleted or it should be clearly stated that the F value was used. Please also indicate the estimated incidence of using q2 in Table 1.

2. Table 2. Estimates of SMA births in UAE and KSA and the associated cost of postnatal treatment versus premarital screening

Is it correct to understand that "Cost reduction" refers to cases where SMA is diagnosed through prenatal diagnosis and an abortion is performed? The journal "communications medicine" is an international magazine, so this assertion by the authors may be difficult for some readers to accept. This is because there are diverse cultures, diverse religions, and diverse ways of thinking. I think it would be better to remove the mention of costs from the abstract and main text, and also remove Table 2. If it could not be deleted, I think that it would be better to add the possibility of ultra-early treatment (treatment just after birth) for SMA.

3. Abstract

"Methods

We preformed SMA diagnostic testing in a clinical multi-national patient cohort (N = 171) referred for hypotonia and/or muscle weakness."

"preformed" might be "performed".

Reviewer #1 (Remarks to the Author):

Dear Editor,

I read with interest the case report entitled (Spinal muscular atrophy genetic epidemiology and the case for premarital 2 genomic screening in Arab populations) reported by Dr. Abou-Tayoun and his colleagues. They did SMA diagnostic testing in a clinical multi-national patient cohort referred for hypotonia and/or muscle weakness. In addition, they carried out genetic newborn screening for SMA on 1,252 healthy Emirati newborns to estimate the carrier frequency and incidence of the disease in the United Arab Emirates.

In line 286 (Al-Jalila Children's is the only tertiary pediatric center in the UAE and is 287 the main referral center for pediatric patients with rare diseases across the UAE), just to modify this sentence as there are 2 other centers dealing with rare diseases cross UAE. Suggest to write:

" Al- Jalila Children's is one of tertiary pediatric centers in the UAE dealing with pediatric patients with rare diseases across the UAE".

Thank you for the comment. We have fixed this statement.

Figure 3, without label of A and B.

This is now added.

In SMA Epidemiology in other Arab populations part. It will be great to document why Saudi Arabia chosen to be compared with UAE.

This is a great point. As we mention in results section "SMA epidemiology in other Arab countries", KSA was the only country where a sizeable sample was used for genetic screening (N = 4,189). We also reviewed studies in other Arab countries, but they all suffer from very low sample sizes, including a Moroccan study (also mentioned in this section) which genetically screened for 150 newborns only.

This study supports the previously reported by Hannah V. Wilson, Flavia M. Cantarutti, Islam Eltantawy, Alia A. Aldhaheeri, Yousuf M. Naqvi, Amrit Sadani and Muna M. Al Safi. In Real-World Evidence – The Clinical Burden of Spinal Muscular Atrophy (SMA) in Abu Dhabi, UAE. In which they reported all things related to SMA considered, the clinical burden of SMA in Abu Dhabi is significant, as reported worldwide. Thus the need for screening and prevention are mandated.

Thank you for bringing this study to our attention and happy to hear it agrees with our conclusions that screening is the most efficient preventive approach for this disease.

Reviewer #2 (Remarks to the Author):

This report is well written and organized. Authors describe epidemiology of SMA in Arab population of UAE. Their conclusion is well supported by statistical analysis. They report that incidence of SMA in UAE is higher due to consanguinity. They state that screening for carrier status may reduce incidence of SMA in population and therefore can have a significant economic impact by reducing healthcare spending on treatment of SMA. I would recommend this manuscript for publication.

Thank you for the encouraging comment.

Reviewer #3 (Remarks to the Author):

The authors conducted two surveys. The first is an investigation of SMN1 and SMN2 copy numbers in pediatric patients with hypotonia or/and muscle weakness in the UAE. The second is a newborn screening survey for SMA in the UAE. There are few similar reports in the UAE, and this is considered to be a valuable report. On the other hand, I have some major concerns about this paper.

1. Estimated incidence

Of the 1,252 newborns, 19 had one copy of SMN1 (1.5%). If 19 newborns are carriers, then $19/1,252 = 1$ in 65.9. As the authors write, the incidence (q^2) of SMA calculated from this is 1 in 17,368 ($65.9 \times 65.9 \times 4$) individuals. The authors used the F value to conclude that it was 1 in 5,990 individuals. This is nearly three times more frequent than q^2 . As an SMA expert, I don't understand this process. Therefore, it is important to explain in an easy-to-understand way to the reader the process and reason why the incidence has tripled by using the F value. "For the first time, we estimate SMA carrier frequency (1.5%) and incidence of the disease (1 in 5,990 live births) in the United Arab Emirates" in the abstract should be revised. I think the incidence of using q^2 should be specified in the abstract. The incidence of using the F value in the abstract should be deleted or it should be clearly stated that the F value was used. Please also indicate the estimated incidence of using q^2 in Table 1.

Thank you for this comment. As we explain in results, we did not use q^2 because our population does not satisfy the 'random mating' condition for Hard-Weinberg equilibrium where q^2 can be applied. As you know, consanguinity rates, including first cousin marriages, are high in our setting and using q^2 would be inaccurate. This is also true for all other recessive diseases in this population where the burden of such diseases increases with consanguinity and inbreeding. Therefore, a better estimate of incidence has to include this factor, namely the coefficient of inbreeding (F), as has been used by other studies (please refer to reference 14). We hope this clarifies the reasoning behind using qXF , and we also hope this is clearly explained in results section (please refer to "SMA epidemiology in the United Arab Emirates" section).

2. Table 2. Estimates of SMA births in UAE and KSA and the associated cost of postnatal treatment versus premarital screening

Is it correct to understand that "Cost reduction" refers to cases where SMA is diagnosed through prenatal diagnosis and an abortion is performed? The journal "communications medicine" is an international magazine, so this assertion by the authors may be difficult for some readers to accept. This is because there are diverse cultures, diverse religions, and diverse ways of thinking. I think it would be better to remove the mention of costs from the abstract and main text, and also remove Table 2. If it could not be deleted, I think that it would be better to add the possibility of ultra-early treatment (treatment just after birth) for SMA.

Thank you. We acknowledge the diversity and the availability of several options. However, we are mainly comparing the "premarital screening" option versus "postnatal treatment". Therefore, "cost reduction" refers to this comparison as mentioned in abstract and in manuscript. Premarital screening is more widely accepted in our setting because of the consanguineous nature of the population and the collective high carrier rate for recessive diseases. While prenatal screening and termination is another possible cost-effective option, it is not widely practiced in this region for several reasons.

3. Abstract

"Methods

We performed SMA diagnostic testing in a clinical multi-national patient cohort (N = 171) referred for hypotonia and/or muscle weakness."

"performed" might be "performed".

Thanks. It is now fixed.

REVIEWERS' COMMENTS:

Reviewer #3 (Remarks to the Author):

I was able to resolve my doubts through the comments of the authors. There are no particular concerns. Reviewers' comments:

Reviewer #2 (Remarks to the Author):

Thank you for the revision and tracked changed word document on which I worked. In the first page, the authors acknowledge that AMR is not only a problem related to overuse of antibiotics, but then already starting at the end of page 1 they quickly revert to focusing on overuse as main component to consider when undergoing economic evaluations (also Table 1 and the elements that pose a challenge). We still do not understand the reason for this focus, please explain or change to optimising use. To note, that it's the latter that will have an impact on AMR, not necessarily only reducing use (this is also acknowledged by the authors' use of reference 10).

Reviewer #3 (Remarks to the Author):

- Generally the manuscript is an important addition to the literature and will help stimulate debate around how to more routinely incorporate cost-effective analyses (CEA) in the evaluation of AMR interventions. I have a few specific points the author may wish to consider:
- It appears that authors focus exclusively on antibiotics, but much of the discussion is relevant to antimicrobials generally, therefore the authors may wish to explicitly state they are only concerned with antibiotics from the beginning, or acknowledge these principles are applicable to antimicrobials
- LMICs- do the authors wish to discuss whether CEAs methods and data collection need to be further developed that are applicable to LMICs? Rather than working on the assumption that CEA conducted in high-income countries will automatically promote availability of stewardship interventions in LMICs. If we are taking a more global perspective, increased attention to establishing what is cost-effective in LMICs for AMR may have a greater impact on AMR.
- The authors also need to acknowledge that estimating NPV in one health system or country context may not be generalisable to other countries, and that there may be benefit in development of the frameworks/methods to deploy this approach which can be populated with more country specific data to support national decision making
- The authors need to discuss some of the potential dangers of this approach, in general I am quite supportive of this, and I do believe there needs to be more routine CEA of AMR interventions, however there is a risk that mandating a specific "threshold" for NPV may steer investment only towards interventions where robust data exists on costs and benefits. In the context of scarcity, there is the risk that policy-makers will use CEA as justification for disinvestment rather than investment.
- In terms of the Figure, the pathways of colonisation, selection, and de novo resistance are sensible. The only point I wish to add is whether authors may wish to discuss whether their proposed approach to CEA is actually fairly conservative, as they do not include many of the broader economic benefits of effective antimicrobials outlined by Rothery et al 2018 ie "STEDI" factors (spectrum, transmission, enablement, diversity and insurance). While originally developed to establish value of

novel antimicrobials, they are also applicable to other AMR interventions. Although the drawback of these factors are that they are often difficult to estimate and include within economic models (without significant uncertainty)